# Beyond Nerve Entrapment: A Narrative Review of Muscle–Tendon Pathologies in Deep Gluteal Syndrome

**DOI:** 10.3390/diagnostics15192531

**Published:** 2025-10-07

**Authors:** Yong Hyun Yoon, Ji Hyo Hwang, Ho won Lee, MinJae Lee, Chanwool Park, Jonghyeok Lee, Seungbeom Kim, JaeYoung Lee, Jeimylo C. de Castro, King Hei Stanley Lam, Teinny Suryadi, Kwan Hyun Youn

**Affiliations:** 1Department of Orthopaedic Surgery, Gangnam Sacred Heart Hospital, Hallym University College of Medicine, Seoul 07441, Republic of Korea; lhwghm@gmail.com; 2Incheon Terminal Orthopedic Surgery Clinic, Incheon 21574, Republic of Korea; mjlee951224@gmail.com (M.L.); humanpcw94@gmail.com (C.P.); 2wo02wo0@naver.com (J.L.); 3International Academy of Regenerative Medicine, Incheon 21574, Republic of Korea; perfectceive@gmail.com (J.L.); stplayer@naver.com (S.K.); 4Board of Clinical Research, The International Association of Musculoskeletal Medicine, Kowloon, Hong Kong; drlamkh@gmail.com; 5Bareun Neurosurgery Clinic, Cheongju-si 28402, Republic of Korea; 6Miso Pain Clinic, Suwon-si 16703, Republic of Korea; 7SMARTMD Center for Non-Surgical Pain Interventions, Adventist University of the Philippines, Silang 4118, Cavite, Philippines; jeidec@yahoo.com.ph; 8The Faculty of Medicine, The University of Hong Kong, Pokfulam, Hong Kong; 9The Faculty of Medicine, The Chinese University of Hong Kong, New Territories, Hong Kong; 10Department of Physical Medicine and Rehabilitation, Hermina Podomoro Hospital, North Jakarta 14350, Indonesia; painfreedoc22@gmail.com; 11Department of Physical Medicine and Rehabilitation, Medistra Hospital, South Jakarta 12950, Indonesia; 12Physical Medicine and Rehabilitation, Synergy Clinic, West Jakarta 11510, Indonesia; 13Division in Biomedical Art, Incheon Catholic University Graduate School, Incheon 06591, Republic of Korea; artanato@naver.com

**Keywords:** deep gluteal syndrome, sciatic nerve entrapment, enthesopathy, piriformis syndrome, ultrasonography, hydrodissection, prolotherapy, diagnostic algorithm

## Abstract

Deep Gluteal Syndrome (DGS) has traditionally been defined as a clinical entity caused by sciatic nerve (SN) entrapment. However, recent anatomical and imaging studies suggest that muscle- and tendon-origin pathologies—including enthesopathy—may also serve as primary pain generators. This narrative review aims to broaden the current understanding of DGS by integrating muscle and tendon pathologies into its diagnostic and therapeutic framework. The literature was selectively reviewed from PubMed, Cochrane Library, Google Scholar, PEDro, and Web of Science to identify clinically relevant studies illustrating evolving concepts in DGS pathophysiology, diagnosis, and management. We review clinical features and diagnostic tools including physical examination, MRI, and dynamic ultrasonography, with special attention to deep external rotator enthesopathy. Treatment strategies are summarized, including conservative therapy, ultrasound-guided injections, hydrodissection, and prolotherapy. This narrative synthesis underscores the importance of recognizing muscle-origin enthesopathy and soft-tissue pathologies as significant contributors to DGS. A pathophysiology-based, multimodal approach is essential for accurate diagnosis and effective treatment.

## 1. Introduction

Sciatica accompanied by leg numbness and motor weakness is typically suspected to result from lumbar pathologies such as lumbar disc herniation or spinal stenosis. However, in clinical practice, some patients continue to experience symptoms despite treatment for these spinal conditions, indicating the presence of non-discogenic pain [1]. Recently, Deep Gluteal Syndrome (DGS) has been gaining attention as one of the major causes of such non-discogenic pain.

DGS describes a clinical condition characterized by buttock and radiating leg pain. This pain is generated by structural or functional abnormalities within the deep gluteal space. Its reported prevalence ranges from 5% to 36% [2], and up to17% of patients with sciatica may actually have DGS [3], underscoring its clinical significance and the risk of delayed or missed diagnosis. The term was first introduced by McCrory and Bell in 1999, originally referring to compression of the sciatic nerve (SN) [4]. However, the deep gluteal space is anatomically complex, containing many muscles (e.g., piriformis, obturator internus), tendons and neurovascular structures (e.g., sciatic nerve, superior/inferior gluteal nerves). This complexity allows for a wide range of pathologies beyond classical sciatic nerve entrapment. According to Koh et al., not only the SN but also the superior/inferior gluteal nerves, pudendal nerve, and posterior femoral cutaneous nerve can be entrapped, contributing to chronic buttock pain. Functional abnormalities such as muscle spasm and pelvic instability are also reported to be part of the pathomechanism [5]. Furthermore, pathologies such as muscle hypertrophy, anatomical variations in the gemelli–obturator internus complex, and hamstring tendinopathy can directly cause pain. They may also secondarily irritate nearby nerves [5]. Thus, DGS pathogenesis should not be limited to SN compression alone, as pathologies in surrounding nerves, muscles, and tendons also play a critical role in symptom generation.

While SN entrapment has traditionally defined DGS, recent insights into muscular and enthesis-related pathologies necessitate a broader narrative re-evaluation. This review aims to contextualize sciatic nerve entrapment within this expanded framework. We emphasize how muscular and tendinous pathologies can act as primary pain generators and propose a diagnostic approach that incorporates these broader considerations.

## 2. Method

### 2.1. Literature Search Strategy

This narrative review was conducted by systematically searching the literature to identify studies relevant to the pathophysiology, diagnosis, and management of Deep Gluteal Syndrome (DGS), with a focus on muscle and tendon pathologies. We systematically searched electronic databases—including PubMed, Cochrane Library, Google Scholar, PEDro, and Web of Science—from their inception until December 2024. The search strategy utilized a combination of keywords and Medical Subject Headings (MeSH) terms such as: ‘deep gluteal syndrome’, ‘sciatic nerve entrapment’, ‘piriformis syndrome’, ‘enthesopathy’, ‘proximal hamstring tendinopathy’, ‘ischiofemoral impingement’, ‘ultrasound-guided injection’, ‘hydrodissection’, and ‘prolotherapy’. Boolean operators (AND, OR) were used to combine terms.

### 2.2. Study Selection and Inclusion Criteria

The literature selection aimed to include clinical studies (randomized controlled trials, cohort studies, case–control studies, case series), review articles, and seminal anatomical studies published in English. Studies were included if they focused on:The anatomy of the deep gluteal space,Diagnostic approaches (e.g., history, physical examination, imaging), orTherapeutic interventions for DGS beyond classic nerve entrapment, particularly those involving muscular or tendinous pathologies.

Exclusion criteria consisted of studies focusing solely on lumbar radiculopathy without discussion of extraspinal causes, non-English publications, and conference abstracts without available full text.

The initial search yielded 2189 records across all databases. After removal of duplicates (*n* = 500) and screening of titles/abstracts, 209 studies were assessed for full-text eligibility. Finally, 59 studies were included in the narrative synthesis (Figure 1).

### 2.3. Data Synthesis and Limitations

Given the narrative nature of this review and the heterogeneity of the available literature—which often consists of small case series and expert opinions—a formal systematic review or meta-analysis was not feasible. Consequently, a standardized risk-of-bias assessment of individual studies was not performed. The results were synthesized qualitatively. Study selection was based on clinical relevance, methodological clarity, contribution to the conceptual expansion of DGS, and peer-reviewed status. This approach, while pragmatic for a narrative review, is a limitation and may introduce selection bias. The findings and recommendations should therefore be interpreted in the context of the evolving evidence base for DGS.

To maximize transparency and reproducibility in our study selection process, we followed the guidance of the Preferred Reporting Items for Systematic Reviews and Meta-Analyses (PRISMA) statement. The flow of information through the different phases of the review is illustrated in Figure 1, which provides a detailed overview of the number of records identified, included, and excluded, along with the reasons for exclusions.

To facilitate critical appraisal, we adopted the Oxford Centre for Evidence-Based Medicine (OCEBM) Levels of Evidence system for key interventions [6]. The Levels of Evidence (LoE) are provided in the table in Section 4.2 for diagnostic tests and in the respective treatment sections. Briefly, LoE 1 represents the highest level of evidence (e.g., systematic reviews of high-quality studies), LoE 4 denotes case-series or case–control studies, and LoE 5 indicates evidence based on expert opinion, physiological reasoning, or preliminary findings. This grading allows clinicians to gauge the strength of the recommendations derived from the available literature.

## 3. Anatomy and Pathogenesis

### 3.1. Traditional View: Sciatic Nerve Compression

The deep gluteal space is anatomically bordered anteriorly by the posterior acetabular column, hip joint capsule, and proximal femur, and posteriorly by the gluteus maximus muscle. Laterally, it is bounded by the gluteal tuberosity and the lateral lip of the linea aspera, while the medial border is defined by the sacrotuberous ligament and falciform fascia. Superiorly, the boundary is formed by the sciatic notch, and inferiorly by the origin of the proximal hamstring tendons at the ischial tuberosity (IT) [7].

This space contains the piriformis (PM), gemelli, obturator internus/externus, and quadratus femoris muscles, as well as the sciatic, superior/inferior gluteal, pudendal, and posterior femoral cutaneous nerves (PFCN), in addition to various vascular structures [7].

The SN, which traverses this space, is typically located an average of 1.2 cm lateral to the IT [8]. It is known to exhibit approximately 28 mm of excursion during hip flexion [9]. Recent studies have reported that during hip flexion combined with abduction and external rotation, the SN glides along the posterior margin of the greater trochanter [7,10]. Moreover, its excursion may be further influenced by the degree of knee flexion [11]. According to Smoll et al., anatomical variation between the SN and PM exists in approximately 16.2% of the population, potentially affecting the nerve’s mobility during hip and knee motion [12]. Thus, understanding the course and dynamic movement of the SN within the deep gluteal space is essential in explaining the typical entrapment mechanism observed in DGS.

### 3.2. Expanding View: Muscle-Origin Pain and Enthesopathy

Deep gluteal syndrome may also result from muscular and tendinous pathology, independent of direct SN entrapment. For instance, ischiofemoral impingement occurs when the space between the ischial tuberosity and the lesser trochanter is abnormally narrowed, compressing adjacent soft tissues such as the quadratus femoris muscle. When this interval narrows to less than approximately 17 mm on MRI, the likelihood of pathological impingement increases [13]. Similarly, chronic inflammation or partial tears of the proximal hamstring tendons—originating at the ischial tuberosity, including the semimembranosus, semitendinosus, and biceps femoris—may affect nearby structures such as the pudendal nerve. This pathology is referred to as proximal hamstring tendinopathy [14,15].

The muscles located within the deep gluteal space include the piriformis (PM), obturator internus (OI), obturator externus (OE), superior gemellus (SG), inferior gemellus (IG), and quadratus femoris (QF) [16,17] (Figure 2). These are collectively referred to as the “deep external rotators,” primarily responsible for external rotation of the hip joint. Their proximal tendons are typically attached to the pelvic bones at anatomical regions known as entheses, which are susceptible to injury through repetitive microtrauma, overuse, or degenerative changes [18].

Initially, the pathologic response involves local inflammation. However, repeated mechanical stress may induce “neurovascular invasion” in the region surrounding the tendon, whereby newly formed blood vessels and nerve fibers infiltrate the enthesis. This process can increase nociceptive sensitivity and potentially exert secondary compression on adjacent neural structures [18]. Notably, when enthesopathy affects deep gluteal muscles such as the PM, OI, or QF, resultant swelling or structural alterations may impinge upon nearby peripheral nerves—including the SN, pudendal nerve, and posterior femoral cutaneous nerve—leading to secondary neuropathic pain [19,20].

Ultimately, it is not only the trajectory of the SN that must be considered when evaluating pain and neurologic symptoms within the deep gluteal space. Pathologies of the deep external rotator muscles and proximal hamstring tendons also constitute significant contributors. With increasing reports of MRI or ultrasonographic findings demonstrating abnormalities in the quadratus femoris or proximal hamstring tendinopathy, it is imperative that DGS be approached from an integrated perspective that includes muscle-tendon pathologies, rather than being viewed solely as a nerve entrapment syndrome [21,22,23].

## 4. Diagnosis

### 4.1. History Taking

In patients presenting with posterior hip pain, it is critical to first rule out lumbar lesions. Sciatica originating from lumbar spine pathology can closely mimic the clinical presentation of Deep Gluteal Syndrome (DGS), necessitating a detailed evaluation of the pain’s onset and characteristics during history taking. In particular, pain that worsens with lumbar flexion or extension, or radicular symptoms that follow a specific dermatome, are indicative of lumbar involvement and warrant further imaging evaluation [24].

By contrast, patients with DGS typically report pain localized to the deep posterior hip region that worsens after prolonged sitting—generally around 20 to 30 min [25]. The pain may radiate to the posterior thigh; however, this radiation is nonspecific and does not follow a clear dermatomal distribution, distinguishing it from classical radiculopathy [24].

In cases of SN entrapment, some patients also report experiencing a “release phenomenon,” which manifests as delayed paresthesia—such as pins and needles—when changing posture or standing up after prolonged seated compression [26]. This phenomenon, characterized by sensory disturbance following the relief of nerve compression, may serve as a diagnostic clue for nerve-related forms of DGS [26].

Not all cases of DGS, however, are due to SN entrapment. Muscular and tendinous pathologies within the deep gluteal space may also cause similar symptoms. For instance, ischiofemoral impingement syndrome often causes pain that is exacerbated during terminal hip extension, such as during long-stride walking, and may be accompanied by a snapping sensation of the hip [7,13]. In contrast, patients with proximal hamstring tendinopathy frequently report focal pain in the lower buttock or around the ischial tuberosity following repetitive or high-intensity activity such as running or jumping [14]. The pain typically worsens during the initial heel-strike phase of gait [19].

### 4.2. Clinical Differentiation Between Nerve and Tendon Causes

In the physical examination of patients presenting with sciatica, the first step is to differentiate lumbar spinal pathology. Lumbar foraminal stenosis, particularly when accompanied by posterior facet joint disease, and dorsolateral disc herniation are among the most common causes of radicular symptoms [27]. These conditions typically produce unilateral radiating leg pain and can closely resemble the clinical presentation of DGS, making differential diagnosis challenging. Therefore, it is clinically important to perform the Straight Leg Raising Test (SLRT), a specialized maneuver used to evaluate whether radiating pain is elicited due to nerve root irritation [28,29].

If the results of the physical examination suggest a low likelihood of lumbar involvement, DGS should be strongly considered. The physical examination for DGS should be conducted in a stepwise manner, including inspection, palpation, passive testing, and resistive testing. This structured approach enables comprehensive assessment of pain characteristics and localization, as well as potential functional abnormalities in muscular and neural structures.

During inspection, the clinician should assess biomechanical axial alignment and pelvic symmetry in the standing position. In the seated position, patients may demonstrate an antalgic posture in an attempt to avoid pain. While walking, attention should be paid to the presence of an antalgic gait, leg length discrepancy, or a positive Trendelenburg sign [17]. In long-standing cases, gluteal atrophy on the affected side may also be observed and should be carefully noted [30].

Palpation is useful in refining the localization of tenderness and identifying specific pathological structures. Tenderness over the sciatic notch may suggest piriformis muscle involvement, while medial tenderness at the ischial tuberosity (IT) may indicate pudendal nerve entrapment. Lateral tenderness over the IT may instead point to proximal hamstring tendinopathy (PHT) or ischiofemoral impingement [31].

Passive tests assess the range and limitation of joint motion and are used to characterize the underlying pathology [26,32]. In the diagnosis of DGS, these tests are particularly helpful in evaluating whether pain is provoked by stretching the piriformis or surrounding tissues. The Seated Piriformis Stretch Test, Freiberg Sign, and Flexion-Adduction-Internal Rotation (FAIR) Test are commonly employed for this purpose (Figure 3) [33,34,35,36].

These tests are designed to reproduce conditions that stretch or compress the SN, thereby allowing the clinician to evaluate both the presence and localization of pain. Notably, when pain occurs at the end range of motion, it may suggest the presence of enthesopathy [26,32].

Resistive testing is a valuable tool for assessing whether pain is provoked during contraction of a specific structure, thereby helping to differentiate lesions of the contractile unit—including muscles, tendons, and entheses [26,37]. The interpretation of findings is generally based on the response to isometric resistance [26]:**Pain and weakness** suggest a partial tear of the muscle or tendon, wherein pain-induced inhibition contributes to decreased strength.**Painless weakness** may indicate either a complete rupture of the muscle or tendon or a neurologic deficit.**Full range of motion with pain** typically indicates tendinous pathology, commonly associated with enthesopathy.**Restricted range of motion with pain during muscle stretch** is suggestive of intrinsic muscular pathology such as a muscle tear or may reflect movement limitation due to protective spasm.

Representative resistive tests include Pace’s Sign, Modified Beatty Sign, and the Active Piriformis Test. Pace’s Sign is performed by applying resistance while the patient actively abducts the hip, assessing for deep buttock pain and potential weakness of the piriformis muscle (PM) [38,39]. The Modified Beatty Sign similarly involves resisted hip abduction on the affected side, serving to identify pain reproduction and PM dysfunction [39]. The Active Piriformis Test involves active abduction of the symptomatic leg against resistance and is used to assess buttock pain and muscle weakness; its underlying mechanism is comparable to that of the Modified Beatty Sign (Figure 4) [33].

For a concise overview of sensitivity and specificity of physical examination maneuvers in differentiating DGS subtypes, please refer to Table 1.

### 4.3. Imaging

#### 4.3.1. Plain Radiographs

In patients presenting with pelvic or hip joint pain, standard radiography of the lumbar spine, pelvis, and hip is usetifying fractures, hip osteoarthritis, or sacroiliitis. It can also assist in detecting or excluding pathological findings such as cam or pincer lesions associated with femoroacetabular impingement (FAI), intrapelvic calcifications, or osteophyte formation. However, plain radiographs have limited value in definitively diagnosing Deep Gluteal Syndrome (DGS). Instead, they serve primarily as a first-line modality for ruling out alternative causes of pelvic and hip pain [45].

#### 4.3.2. CT

Pelvic and hip computed tomography (CT) is primarily utilized to exclude alternative causes of hip pain such as femoral head avascular necrosis (AVN) or osseous lesions. Due to its limited soft tissue resolution, CT is not suitable for confirming a diagnosis of Deep Gluteal Syndrome (DGS); however, it remains valuable for detailed evaluation of bony structural abnormalities. For instance, in cases where ischiofemoral impingement—a potential etiologic factor in DGS—is suspected, axial CT images can be used to assess the bony interval and morphological alterations between the ischial tuberosity and the greater trochanter, allowing for identification of structural abnormalities associated with this condition [13].

Additionally, in patients with metallic implants following hip surgery, where magnetic resonance imaging (MRI) may be compromised due to artifact, CT serves as a useful alternative. Therefore, in suspected DGS cases, CT can be employed to quantitatively assess bony abnormalities, while MRI remains essential for evaluating soft tissue structures, thereby enhancing the diagnostic accuracy through a complementary multimodal approach.

#### 4.3.3. MRI

Magnetic resonance imaging (MRI) is the most useful imaging modality for the diagnosis of Deep Gluteal Syndrome (DGS), particularly when performed using a high-resolution 3-Tesla scanner, which allows for detailed assessment of the SN and surrounding structures within the deep gluteal space [19]. In cases where the SN is compressed by a fibrous band, hypertrophied muscle, or posterior joint capsule, findings such as nerve thickening, high signal intensity on T2-weighted images, and alterations in nerve course may indicate entrapment [5,46]. Furthermore, MR neurography enhances the visualization of the nerve in relation to adjacent tissues, facilitating the identification of abnormal nerve trajectories or severe adhesions [46,47].

However, DGS may also result from intrinsic muscular or tendinous pathologies within the deep gluteal space. Tendon and muscle abnormalities can be evaluated by identifying peritendinous fat plane loss on T1-weighted images and increased intratendinous signal intensity and peritendinous edema on T2-weighted sequences. Tendon thickening and partial tearing may also be observed (Figure 5) [19]. Representative examples include quadratus femoris abnormalities due to ischiofemoral impingement (Figure 6) and proximal hamstring tendinopathy. Nevertheless, due to the anatomical depth and overlapping with adjacent structures, lesions involving the deep external rotators—including the piriformis, gemelli, obturator internus/externus, and quadratus femoris—are often difficult to visualize on MRI, posing diagnostic limitations.

Therefore, imaging evaluation of DGS should include not only assessment for nerve entrapment but also detailed analysis of muscle and tendon abnormalities within the deep gluteal space.

#### 4.3.4. Ultrasound

Ultrasonography has traditionally been used as an adjunctive modality in the diagnosis of Deep Gluteal Syndrome (DGS), but recent studies have increasingly recognized it as a reliable diagnostic tool [17]. One of its key advantages lies in the ability to visualize periarticular soft tissues of the hip—such as muscles, tendons, and nerves—in real time.

According to a study published in 2015 [21], dynamic ultrasonography enables comprehensive evaluation of not only the piriformis muscle but also the broader deep external rotator musculature. Long-axis imaging further enhances diagnostic value. By dynamically assessing the relative motion between the muscles and the SN during internal and external rotation of the hip, ultrasonography facilitates clearer identification of pathological mechanisms involved in conditions such as piriformis syndrome, obturator–gemelli complex lesions, and ischiofemoral impingement.

While MRI lacks sensitivity and specificity for detecting peripheral enthesitis, ultrasonography may offer superior sensitivity and greater accessibility in the evaluation of enthesopathic changes [48,49]. Using both grey-scale and power Doppler techniques, ultrasonography allows for concurrent assessment of structural abnormalities (e.g., tendon thickening, hypoechogenicity, enthesophytes, erosions) and active inflammation (increased vascularity) [49]. This capacity makes it particularly effective for detecting enthesopathic changes that may be overlooked on MRI (Figure 7).

Given its lack of radiation exposure, relatively low cost, and potential for complementary use alongside other imaging modalities, ultrasonography represents a valuable diagnostic tool in patients with suspected DGS.

Figure 8 summarizes the proposed diagnostic algorithm, outlining key steps for evaluating DGS from history taking to confirmatory imaging.

## 5. Therapeutic Consideration

Comparative effectiveness between these treatment modalities is not yet well-established due to a lack of head-to-head trials. Generally, a step-up approach is recommended, initiating with conservative management for all patients (LoE: 5). For those failing conservative care, ultrasound-guided injections may be considered before pursuing more invasive options like hydrodissection or surgery. The choice between hydrodissection (potentially more suited for cases with significant neural adhesion) and prolotherapy (potentially targeting enthesopathy more directly) is based on the suspected primary pain generator, though robust comparative evidence is lacking (LoE: 5). Surgical intervention is typically reserved for refractory cases with clear structural pathologies amenable to correction.

The treatment of Deep Gluteal Syndrome (DGS) is based primarily on conservative management, with stepwise escalation to injection therapy or surgical intervention as needed. To date, there is no universally established standard treatment protocol for DGS. Therefore, a personalized therapeutic approach is required, taking into account factors such as symptom duration, pain intensity, and the anatomical etiology of nerve compression.

### 5.1. Conservative Treatment

Initial management of Deep Gluteal Syndrome (DGS) typically begins with conservative treatment. This includes pharmacologic therapy such as nonsteroidal anti-inflammatory drugs (NSAIDs) and muscle relaxants, along with activity modification and physical therapy aimed at improving flexibility and strengthening the deep external rotator muscles [45].

Exercise-based interventions, including stretching and neural mobilization, are useful in alleviating neural compression and muscular dysfunction. Among these, piriformis stretching has long been regarded as a classical non-surgical therapeutic approach [50,51].

Radial extracorporeal shockwave therapy (rESWT) has also demonstrated efficacy in patients with piriformis syndrome, showing improvements in clinical symptoms and piriformis muscle stiffness, as well as a reduction in the cross-sectional area of the SN [52].

However, it is crucial to critically appraise the evidence supporting these conservative measures. While widely recommended, direct comparisons between specific exercise protocols (e.g., neural mobilization vs. piriformis stretching vs. strengthening) are lacking. The existing evidence is largely derived from low-level studies, clinical experience, and extrapolation from other musculoskeletal conditions, highlighting a significant gap in the DGS literature and the need for more structured trials.

### 5.2. Injection Therapy

In cases where conservative treatment fails to provide sufficient symptom relief, ultrasound-guided injection therapy may be considered (LoE: 4). By utilizing ultrasonography to accurately target deep external rotator muscles such as the piriformis, obturator internus, and quadratus femoris, pharmacologic agents can be precisely delivered to reduce muscular tension and inflammation.

**Steroid or local anesthetic injection:** These agents are injected either intramuscularly or perineurally to reduce inflammation.**Botulinum toxin injection:** Particularly effective in piriformis syndrome, botulinum toxin serves to relax hypertonic muscles and alleviate pain. However, a critical appraisal of the literature reveals that evidence is limited to small, unblinded studies and case series (LoE: 4) with a high risk of performance and detection bias [53]. Systematic reviews have highlighted the inconsistency in results across studies and the lack of long-term follow-up data. The therapeutic effects are also temporary, necessitating repeated injections. Consequently, it should be considered an option only after more conventional therapies have failed, with patients counseled about the limited and temporary nature of the evidence supporting its use.

In addition, the use of non-conventional agents such as thiocolchicoside and colchicine has been attempted in some cases; however, the supporting evidence for their efficacy remains limited [53].

**Ultrasound-Guided Hydrodissection:** Recently, ultrasound-guided hydrodissection and prolotherapy have gained attention as novel non-surgical treatment options for Deep Gluteal Syndrome (DGS). Hydrodissection involves the mechanical separation of perineural adhesions using an injected solution composed of 5% dextrose, lidocaine, and betamethasone, thereby restoring nerve mobility. In a study by Yen et al. (LoE: 4), 53 patients with DGS underwent ultrasound-guided SN hydrodissection; among them, 73.6% reported ≥50% pain reduction at 1 week post-injection, and 62.3% maintained this improvement at final follow-up [54]. While these results are promising, it is critical to note the limitations inherent to this level of evidence. As a single-arm case series, the study lacks a control group for comparison, making it difficult to attribute the improvements solely to hydrodissection versus placebo effect or natural history of the condition. Additionally, the mid-term follow-up period and relatively small sample size limit the generalizability and strength of these findings. Confirmation through larger, randomized controlled trials with sham-procedure control groups is necessary to firmly establish efficacy.

Notably, hydrodissection differs from conventional injection therapies in that it not only facilitates nerve release, but also directly targets enthesopathic lesions in the deep external rotators—such as the piriformis, obturator internus, and quadratus femoris muscles. Therefore, ultrasound-guided hydrodissection may serve as a comprehensive therapeutic strategy in DGS, simultaneously addressing both muscle-origin enthesopathy and nerve entrapment.

**Prolotherapy:** Prolotherapy is a regenerative injection technique involving the administration of hypertonic dextrose solutions into lax ligaments and enthesopathic regions to stimulate tissue repair [55]. First introduced by Hackett, prolotherapy is now widely applied in a variety of musculoskeletal conditions, including greater trochanteric pain syndrome (GTPS) and knee osteoarthritis [56]. In cases of DGS arising from enthesopathy, ultrasound-guided prolotherapy precisely delivered to the pathological lesion may yield favorable therapeutic outcomes. However, the current evidence is primarily based on anecdotal reports and small case series (LoE: 4) [57] with heterogeneous methodologies and outcome measures. A significant limitation is the absence of high-quality randomized controlled trials comparing prolotherapy to other standard interventions or placebo. Furthermore, the precise mechanism of action for prolotherapy in enthesopathies remains incompletely understood, and the technique is highly operator-dependent. Therefore, these preliminary findings, while encouraging, must be interpreted with caution. The current body of evidence for prolotherapy in DGS is significantly limited by the absence of randomized controlled trials, small sample sizes in existing case series, heterogeneity in injection protocols and outcome measures, and a lack of long-term follow-up data. Furthermore, the operator-dependent nature of the procedure introduces another variable affecting reproducibility and outcomes. Consequently, prolotherapy should currently be considered an investigational treatment option for DGS, to be used within the context of clinical studies or after exhausting conventional evidence-based treatments, with patients being fully informed of the experimental nature of the intervention. Collectively, while these injection therapies show promise, the current body of evidence is characterized by a high risk of bias, heterogeneity in techniques and outcome measures, and a lack of long-term follow-up data. The promising results from case series (LoE 4) must be interpreted with caution until confirmed by well-designed randomized controlled trials with sham-procedure control groups and standardized protocols.

Although multiple interventional options have been described, direct head-to-head comparisons are lacking. Steroid and local anesthetic injections remain widely used but offer only temporary relief. Botulinum toxin may be particularly effective in piriformis-related spasm, but current evidence is limited to small uncontrolled series. Hydrodissection appears promising for nerve-dominant cases, restoring sciatic nerve mobility, whereas prolotherapy may be more suited for tendon-dominant DGS by targeting enthesopathic lesions. However, no randomized trials directly compare these interventions, and long-term outcome data are sparse. This highlights the need for comparative effectiveness studies to determine the most appropriate therapy for each DGS subtype.

**Clinical implications**. The expanding view of DGS has direct consequences for clinical practice. Patients should no longer be evaluated solely for sciatic nerve entrapment, but also for tendon- and muscle-related lesions that may mimic or overlap with neural symptoms. In nerve-dominant DGS, ultrasound-guided hydrodissection or nerve-focused interventions may be prioritized, whereas tendon-dominant cases may benefit more from prolotherapy or structured rehabilitation programs. Mixed presentations often require a multimodal approach, progressing from conservative therapy to targeted injections and, if necessary, surgical decompression. Incorporating patient history (e.g., intolerance to prolonged sitting vs. focal ischial pain), physical examination maneuvers, and complementary imaging (MRI neurography for nerve involvement, ultrasonography for enthesopathy) into diagnostic algorithms allows clinicians to tailor management to the dominant mechanism.

### 5.3. Role of Surgical Options

In patients with Deep Gluteal Syndrome (DGS) who do not respond to conservative management for more than three months, surgical intervention—either endoscopic or open—may be considered [58]. The choice of surgical approach depends on the type and etiology of the lesion. In particular, resection of space-occupying lesions that compress the nerve may be necessary in cases involving suspected malignancy or chronic neuropathic deficits [17].

Endoscopic SN decompression has been reported as a minimally invasive technique that allows direct visualization of entrapment sites within the deep gluteal space, enabling removal of fibrous bands or pathological adhesions [3,59].

Various surgical techniques may be applied depending on the specific pathology: resection of the lesser trochanter in cases of refractory ischiofemoral impingement (IFI), release of the piriformis muscle in piriformis syndrome (PS), and endoscopic SN neurolysis in persistent DGS [3,60].

Finally, we have presented a algorithm chart illustrating how to treat deep gluteal syndrome (Figure 9).

## 6. Conclusions

This narrative review highlights the need to expand the diagnostic and therapeutic approach to DGS beyond SN entrapment. By integrating muscle- and tendon-origin pathologies, clinicians can better explain cases of posterior hip pain that are unresponsive to spine-based treatments.

Recognizing the diverse pathomechanisms underlying DGS is essential for establishing an accurate diagnosis and devising an effective, individualized treatment strategy. Given the potential for overlapping clinical presentations between neural and soft tissue abnormalities within the deep gluteal space, a comprehensive approach is warranted. This includes detailed history taking, thorough physical examination, high-resolution MRI, and dynamic ultrasonography.

In addition, emerging technologies such as ultrasound-guided hydrodissection—which can simultaneously target nerve entrapment and enthesopathic lesions—reflect an expanded clinical understanding of DGS and are gaining attention as integrative treatment modalities. Future research should not only focus on validating novel therapies through robust RCTs but must also prioritize the development and validation of a practical, pathoanatomy-based classification system for DGS (e.g., nerve-dominant vs. tendon-dominant vs. mixed). This system should be based on reproducible clinical criteria (e.g., specific pain patterns from history and combinations of provocative tests from Table 1) and imaging biomarkers (e.g., MRI neurography for nerve-dominant DGS and ultrasound-visualized enthesopathy for tendon-dominant DGS). Subsequently, this classification must be validated in prospective cohorts and tested in treatment-stratified trials (e.g., comparing nerve hydrodissection in nerve-dominant DGS vs. prolotherapy in tendon-dominant DGS) to establish Level 1 evidence for personalized treatment algorithms. This stepwise approach is crucial for moving DGS management from a generalized framework to a precision medicine framework.

Furthermore, we have provided Levels of Evidence (LoE) based on the Oxford Centre for Evidence-Based Medicine framework for the key therapeutic interventions discussed. This highlights that the current management of DGS, particularly novel techniques like hydrodissection and prolotherapy, is often supported by lower-level evidence (LoE 4–5), underscoring the critical need for more high-quality randomized controlled trials (LoE 1–2) in the future.

## Figures and Tables

**Figure 1 diagnostics-15-02531-f001:**
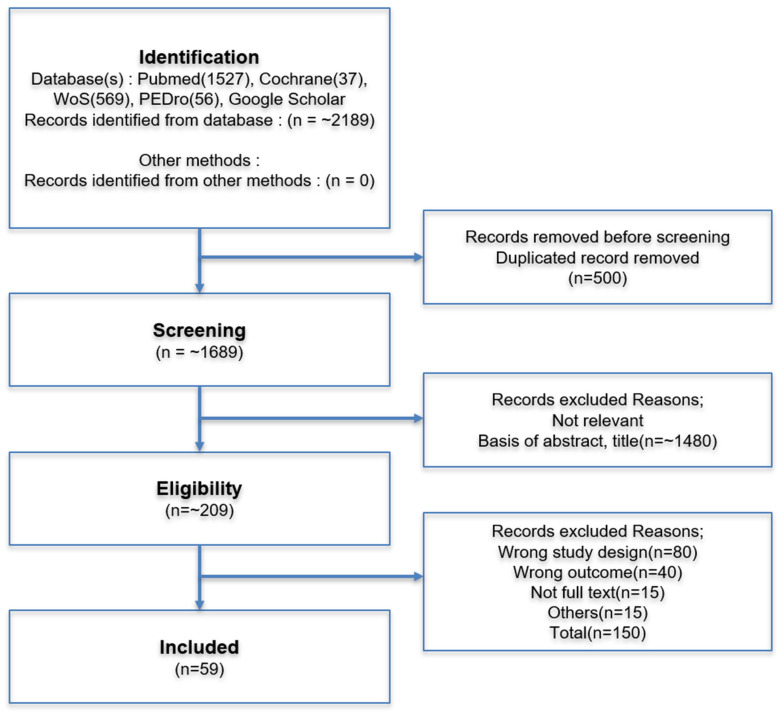
Flow diagram of the study selection process for the narrative review.

**Figure 2 diagnostics-15-02531-f002:**
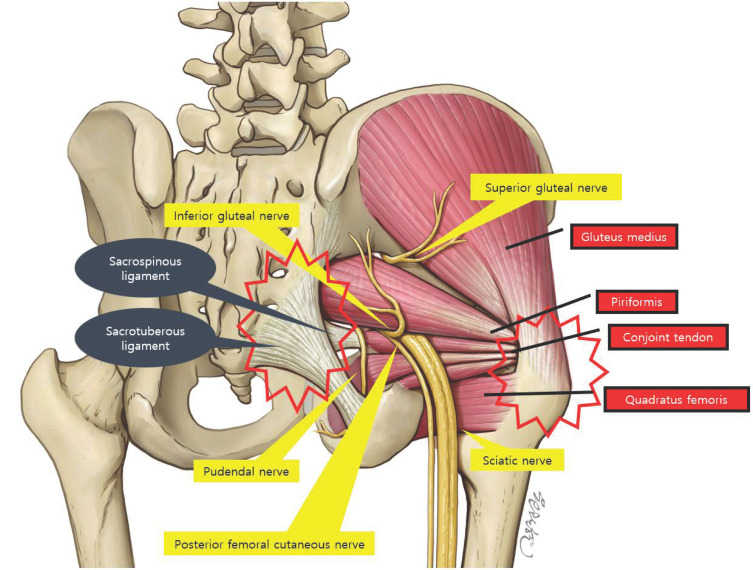
Anatomy of the deep gluteal space. Muscles and ligaments are indicated in black, and nerves are indicated in yellow boxes. The area with red stars is where enthesopathy occurs. Ligaments are indicated by black circles, nerves by yellow boxes, and tendons by red boxes.

**Figure 3 diagnostics-15-02531-f003:**
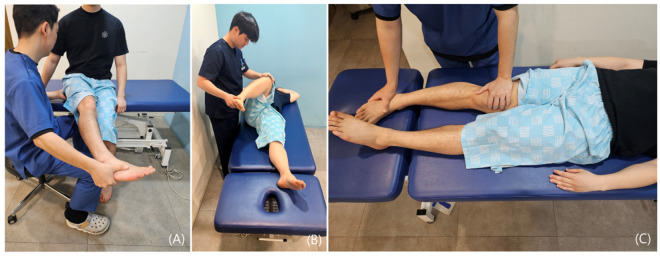
(**A**) *Seated Piriformis Stretch Test*: This test is performed with the patient seated at the edge of the examination table, with the hip flexed to approximately 90 degrees and the knee extended, in order to elongate the PM. While palpating the sciatic notch, the examiner passively adducts and internally rotates the patient’s leg to provoke symptoms. (**B**) *Flexion Adduction Internal Rotation (FAIR) Test*: This maneuver involves flexion, adduction, and internal rotation of the hip to assess compression of the SN. (**C**) *Freiberg Sign*: With the leg in extension, this test involves passive internal rotation of the hip and is used to evaluate pain provocation and range of motion in the hip joint. All photographs were taken by the authors for the present study.

**Figure 4 diagnostics-15-02531-f004:**
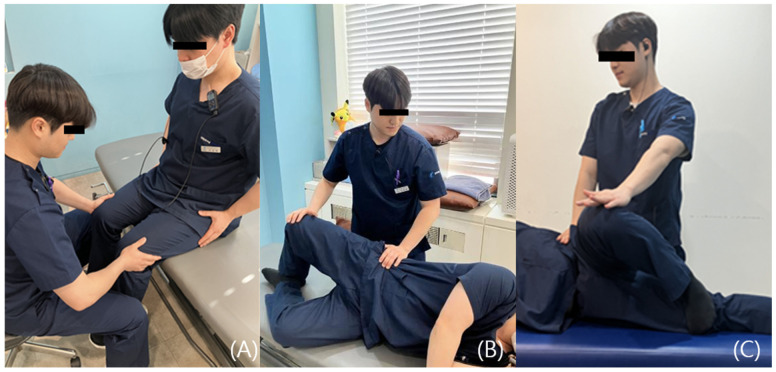
(**A**) Pace’s Sign: With the patient seated on the examination table and the thighs positioned in a relaxed, neutral alignment, the examiner places both hands on the lateral aspects of the patient’s knees and applies adduction resistance. The patient is then instructed to actively abduct the hips. The test is considered positive if buttock pain is reproduced or if there is noticeable weakness of the piriformis muscle. (**B**) Active Piriformis Test: In the side-lying position, the patient positions the affected lower limb uppermost with the knee flexed. The foot is placed on the table behind the unaffected limb. While palpating the piriformis muscle, the examiner places a hand on the lateral aspect of the patient’s upper knee and applies resistance to adduction and internal rotation. The patient is instructed to maintain contact between the heel and the table while actively abducting and externally rotating the hip. (**C**) Modified Beatty Test: In the side-lying position, the unaffected lower limb is kept fully extended at the hip and knee. The affected limb is flexed to 90 degrees at both the hip and the knee. The patient then performs active hip abduction while the examiner applies adduction resistance. The test is positive if the maneuver reproduces buttock pain or highlights muscular weakness. All photographs were taken by the authors for the present study.

**Figure 5 diagnostics-15-02531-f005:**
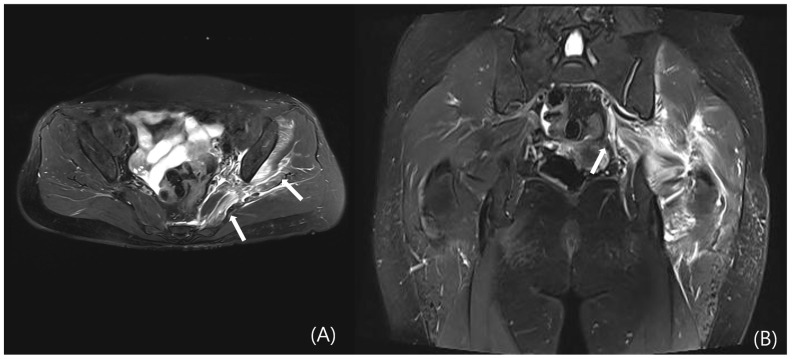
Contrast-enhanced pelvic MRI of a patient who presented with left buttock pain and fever that began two days prior to admission. At the time of presentation, no erythema or warmth was noted over the buttock area, but tenderness was observed. (**A**) Axial and (**B**) coronal T2-weighted images reveal diffuse enlargement and signal intensity changes with contrast enhancement in the left piriformis, gluteus minimus, and obturator internus muscles (white arrows).

**Figure 6 diagnostics-15-02531-f006:**
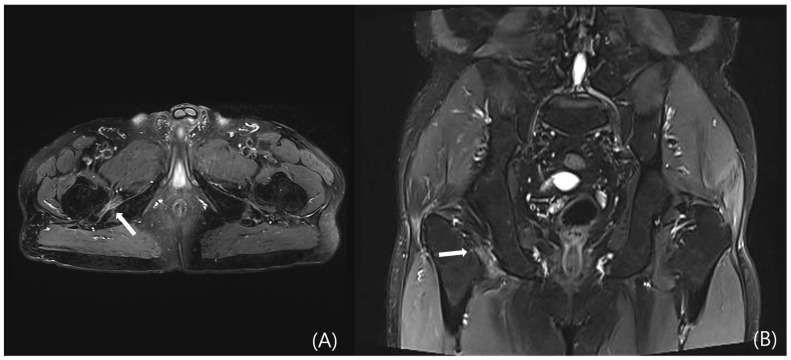
Pelvic MRI of a patient who presented with right posterior hip pain and radiating pain to the right thigh, which had developed five months prior to admission. The patient reported characteristic worsening of pain after walking for more than 20 min. (**A**) Axial and (**B**) coronal T2-weighted images demonstrate edema of the right quadratus femoris muscle, associated with narrowing of the ischiofemoral (IF) space (white arrows).

**Figure 7 diagnostics-15-02531-f007:**
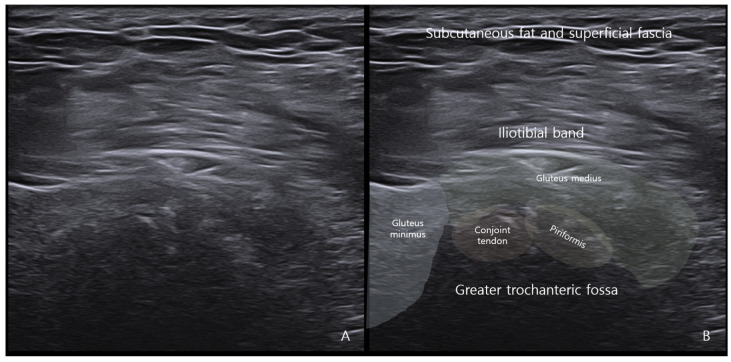
A case of deep gluteal syndrome diagnosed by ultrasound. (**A**) Calcification is observed at the point where the piriformis conjoint tendon attaches to the greater trochanteric fossa. (**B**) Structures attached to the greater trochanter were schematically and named using ultrasound.

**Figure 8 diagnostics-15-02531-f008:**
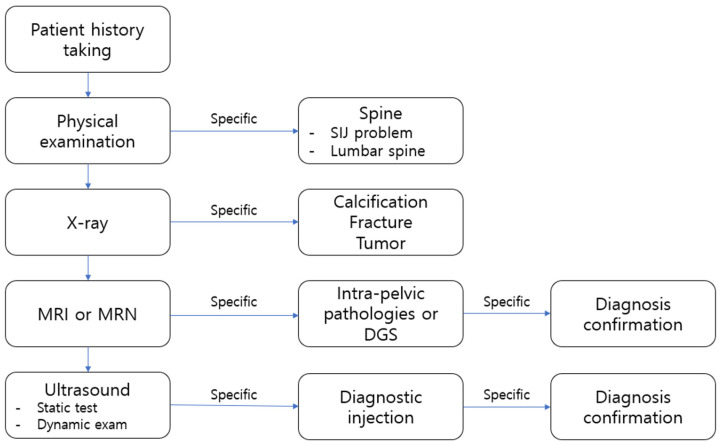
A diagrammatic algorithm chart for diagnosing deep gluteal syndrome is shown.

**Figure 9 diagnostics-15-02531-f009:**
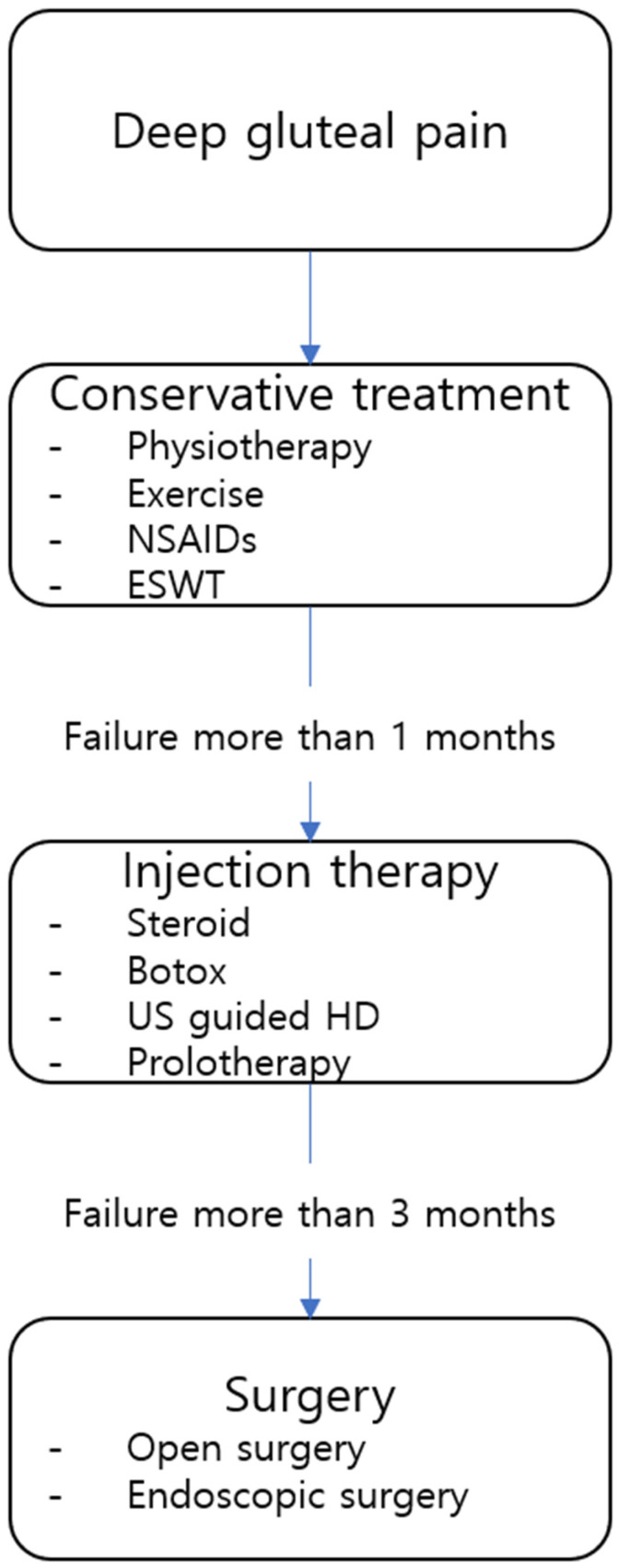
A diagrammatic algorithm chart for treatment DGS is shown.

**Table 1 diagnostics-15-02531-t001:** Summary of Clinical Tests and Diagnostic Performance in DGS Evaluation.

No.	Clinical Tests	Clinical Subtype	Sensitivity	Specificity	LoE	Title	Reference	Year	Journal
**1**	straight leg test (SLRT)	discogenic radiculopathy	28% (95% CI: 22 to 35)	90% (95% CI: 85 to 94)	1a	Physical examination for lumbar radiculopathy due to disc herniation in patients with low-back pain	[40]	2010	Cochrane Database Syst. Rev.
**2**	straight leg test (SLRT)	discogenic radiculopathy	0.15 (0.05 to 0.33)	0.95 (0.68 to 1.00)	4	Diagnostic accuracy of clinical tests for sciatic nerve entrapment in the gluteal region	[33]	2014	Knee Surg. Sports Traumatol. Arthrosc.
**3**	straight leg test (SLRT)	discogenic radiculopathy	62.24% (56.75–67.72)	45.91% (38.66–53.20)	4	Sensitivity and Specificity of Modified Bragard Test in Patients with Lumbosacral Radiculopathy Using Electrodiagnosis as a Reference Standard	[41]	2018	J. Chiropr. Med.
**4**	Modified Straight Leg Raise Test (mSLRT)	discogenic radiculopathy	Not reported	Not reported	5	Frequently described in clinical practice, but no validated diagnostic accuracy studies are available; primarily supported by expert opinion.			
**5**	Bowstring Test	discogenic radiculopathy	Not reported	Not reported	5	Historically reported in the orthopedic literature; however, no standardized sensitivity or specificity data exist, and evidence is limited to descriptive accounts.			
**6**	Seated piriformis stretching	piriformis	0.52 (0.33 to 0.71)	0.90 (0.60 to 0.98)	4	Diagnostic accuracy of clinical tests for sciatic nerve entrapment in the gluteal region	[33]	2014	Knee Surg. Sports Traumatol. Arthrosc.
**7**	Seated piriformis stretching	piriformis	91%	80%	5	Hip Pain in Adults: Evaluation and Differential Diagnosis	[42]	2021	Am. Fam. Physician
**8**	Seated piriformis stretching	piriformis	52%	90%	5	Deep Gluteal Syndrome: A Pain in the Buttock	[16]	2021	Curr. Sports Med. Rep.
**9**	Freiberg Sign	piriformis	Not reported	Not reported	5	-Commonly used to assess piriformis-related pain yet lacks quantitative validation; diagnostic accuracy remains based on anecdotal and expert reports. (LoE: 5)			
**10**	Flexion Adduction Internal Rotation (FAIR)	piriformis, Gemelli-Obturator Internus Complex	69.6% (372)	59.9% (147)	4	Piriformis syndrome: diagnosis, treatment, and outcome—a 10-year study	[43]	2002	Arch. Phys. Med. Rehabil.
**11**	Flexion Adduction Internal Rotation (FAIR)	piriformis, Gemelli-Obturator Internus Complex	59% to 100%	4% to 75%	5	Hip Pain in Adults: Evaluation and Differential Diagnosis	[42]	2021	Am. Fam. Physician
**12**	Flexion Adduction Internal Rotation (FAIR)	piriformis, Gemelli-Obturator Internus Complex	88.10%	83.20%	5	Deep Gluteal Syndrome: A Pain in the Buttock	[16]	2021	Curr. Sports Med. Rep.
**13**	Flexion Adduction Internal Rotation (FAIR)	piriformis, Gemelli-Obturator Internus Complex	85.00%	82.00%	5	Piriformis syndrome, diagnosis and treatment	[44]	2009	Muscle Nerve
**14**	Beatty sign	piriformis	Not reported	Not reported	5	-Widely cited in textbooks and case reports, but no formal studies evaluating sensitivity or specificity; current use relies on clinical tradition rather than validated evidence. (LoE: 5)			
**15**	Pace’s test	piriformis	Not reported	Not reported	5	Frequently performed in clinical settings to evaluate piriformis involvement; nonetheless, diagnostic performance has not been systematically validated and rests on expert opinion. (LoE: 5)			
**16**	Modified Beatty Sign	piriformis	Not reported	Not reported	5	Proposed as a variant of the Beatty test; described in limited clinical reports without supporting accuracy studies; evidence remains anecdotal. (LoE: 5)			
**17**	Active piriformis test	piriformis	0.78 (0.58 to 0.90)	0.80 (0.49 to 0.94)	4	Diagnostic accuracy of clinical tests for sciatic nerve entrapment in the gluteal region	[33]	2014	Knee Surg. Sports Traumatol. Arthrosc.

Abbreviation: LoE, Level of Evidence according to the Oxford Centre for Evidence-Based Medicine (OCEBM) 2011 guidelines [6] OCEBM Levels of Evidence Working Group. “The Oxford 2011 Levels of Evidence”. Oxford Centre for Evidence-Based Medicine. http://www.cebm.net/index.aspx?o=5653 (accessed on 6 September 2025).

## Data Availability

No new data were created or analyzed in this study. Data sharing is not applicable to this article.

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
