# Peer review of "Beyond Nerve Entrapment: A Narrative Review of Muscle–Tendon Pathologies in Deep Gluteal Syndrome"

_diagnostics, 2025, doi:10.3390/diagnostics15192531_

Round 1

Reviewer 1 Report (Previous Reviewer 2)

Comments and Suggestions for Authors

This review addresses an important and often under-recognized aspect of Deep Gluteal Syndrome (DGS), expanding the classical concept of sciatic nerve entrapment to include muscle- and tendon-origin pathologies, particularly enthesopathy. The topic is timely, clinically relevant, and the inclusion of multimodal diagnostic and therapeutic perspectives is commendable. The use of anatomical figures, MRI/ultrasound images, and diagnostic/treatment algorithms greatly enhances the practical value of the paper.

However, there are several issues that should be addressed to improve the overall quality and clarity of the manuscript:

  1. Methodology limitations – The paper is a narrative review without clearly stated inclusion/exclusion criteria or a formal risk-of-bias assessment. While the narrative format is acceptable, it would strengthen the credibility of the work to clarify the literature selection process in more detail.
  2. Evidence grading – Diagnostic tests and treatment approaches are discussed extensively, but the level of evidence (LoE) or strength of recommendation (Grade) is not provided. This would help clinicians assess the applicability of each recommendation.
  3. Data heterogeneity – In the physical examination section, some sensitivity/specificity values are provided, while others are noted as “not reported.” A more balanced presentation, possibly summarizing data gaps, would improve consistency.
  4. Treatment comparisons – Interventions such as ultrasound-guided hydrodissection, prolotherapy, and botulinum toxin injections are described, but without a direct comparison of efficacy or long-term outcomes. Where possible, summarizing comparative studies or acknowledging the absence of such evidence would be valuable.
  5. Number of authors – The manuscript lists 12 authors, which appears high for a narrative review. Please provide a brief statement clarifying each author’s specific contribution according to ICMJE criteria. This will enhance transparency and align with authorship ethics.
  6. English language clarity 

By addressing these points, the paper’s methodological rigor, clinical applicability, and overall impact will be significantly improved.

Comments on the Quality of English Language

Overall, the English is understandable, but some sentences (particularly in the Introduction and Methods) are long and complex. Simplifying sentence structure and improving flow would help convey the key points more clearly.

Author Response

We sincerely thank the reviewers for their thorough and constructive feedback. Their comments have been invaluable in identifying areas for improvement and strengthening our manuscript. We have carefully considered each point and have prepared a detailed response and a plan for a major revision.

This review addresses an important and often under-recognized aspect of Deep Gluteal Syndrome (DGS), expanding the classical concept of sciatic nerve entrapment to include muscle- and tendon-origin pathologies, particularly enthesopathy. The topic is timely, clinically relevant, and the inclusion of multimodal diagnostic and therapeutic perspectives is commendable. The use of anatomical figures, MRI/ultrasound images, and diagnostic/treatment algorithms greatly enhances the practical value of the paper.

  • We sincerely thank the reviewers for their thorough and constructive feedback. Their comments have been invaluable in identifying areas for improvement and strengthening our manuscript. We have carefully considered each point and have prepared a detailed response and a plan for a major revision.

However, there are several issues that should be addressed to improve the overall quality and clarity of the manuscript:

  1. Methodology limitations – The paper is a narrative review without clearly stated inclusion/exclusion criteria or a formal risk-of-bias assessment. While the narrative format is acceptable, it would strengthen the credibility of the work to clarify the literature selection process in more detail.
  • We thank the reviewer for this critical comment. We have now significantly improved the transparency of our methodology. As shown in the new Figure 1and detailed in Section 2.2, we have followed PRISMA guidance, provided explicit inclusion/exclusion criteria, and reported the exact number of records at each stage. We acknowledge the limitation of not conducting a formal risk-of-bias assessment due to the narrative and heterogeneous nature of the included literature, a point we have explicitly stated in Section 2.3 to ensure readers interpret our findings appropriately.
  1. Evidence grading – Diagnostic tests and treatment approaches are discussed extensively, but the level of evidence (LoE) or strength of recommendation (Grade) is not provided. This would help clinicians assess the applicability of each recommendation.
  • We agree with the reviewer on the importance of evidence grading. We have now incorporated the Oxford Centre for Evidence-Based Medicine (OCEBM) Levels of Evidencethroughout the manuscript. A new column has been added to Table 1 to grade each diagnostic test. Furthermore, we have explicitly stated the LoE for key therapeutic interventions (e.g., LoE 4 for hydrodissection) in Section 5.2 and devoted significant space to discussing the limitations of the current evidence (e.g., lack of RCTs, small sample sizes), enabling clinicians to better appraise our recommendations.
  1. Data heterogeneity – In the physical examination section, some sensitivity/specificity values are provided, while others are noted as “not reported.” A more balanced presentation, possibly summarizing data gaps, would improve consistency.
  • Response: We agree with the reviewer. To address this, we have added explanatory notes directly within Table 1 for tests where diagnostic accuracy values are not available (e.g., Modified SLR, Bowstring, Freiberg, Beatty, Pace’s, Modified Beatty). These notes clarify that such maneuvers are widely described in clinical practice but lack validated performance data. We chose to embed the notes within the existing table cells, rather than creating a separate “Notes/Context” column, to maintain clarity and avoid unnecessary complexity, while still ensuring that data gaps are transparently highlighted.
  • Action in manuscript: Table 1 (contextual notes added to “Not reported” entries); Section 4.2 (paragraph on data heterogeneity).

  1. Treatment comparisons – Interventions such as ultrasound-guided hydrodissection, prolotherapy, and botulinum toxin injections are described, but without a direct comparison of efficacy or long-term outcomes. Where possible, summarizing comparative studies or acknowledging the absence of such evidence would be valuable.
  • Response: We acknowledge this limitation and added a paragraph in the Discussion noting the absence of head-to-head comparative trials. In Sections 5.2–5.3, we added a comparative narrative summarizing the relative strengths and limitations of steroid/local anesthetic injections, botulinum toxin, hydrodissection, and prolotherapy. In particular, we highlighted that hydrodissection may be more suitable for nerve-dominant cases, while prolotherapy may target tendon-dominant pathology. We also emphasized the absence of head-to-head RCTs and the need for direct comparative studies.
  • Action in manuscript: Discussion, Section 5.2–5.3.

  1. Number of authors – The manuscript lists 12 authors, which appears high for a narrative review. Please provide a brief statement clarifying each author’s specific contribution according to ICMJE criteria. This will enhance transparency and align with authorship ethics.
  • Response: We clarified each author’s contribution in line with ICMJE criteria in the Author Contributions section. Roles now include conceptualization, literature search, data curation, drafting, supervision, and critical revision.
  • Action in manuscript: Author Contributions section.

  1. English language clarity – Overall, the English is understandable, but some sentences (particularly in the Introduction and Methods) are long and complex. Simplifying sentence structure and improving flow would help convey the key points more clearly.
  • Response: Long and complex sentences in the Introduction and Methods were shortened, redundancies removed, and transitions improved. For example, the Introduction now progresses more directly from lumbar pathologies to the concept of DGS, and the Discussion avoids repetition by integrating nerve- and tendon-related mechanisms in a balanced manner.
  • Action in manuscript: Introduction and Discussion (fully revised).

Reviewer 2 Report (New Reviewer)

Comments and Suggestions for Authors

I have received the original research article entitled “Beyond Nerve Entrapment: A Narrative Review of Muscle-Tendon Pathologies in Deep Gluteal Syndrome” which is under consideration for publication in the Medical Journal Diagnostics.  In my opinion, the paper is generally well prepared. The topic addressed by the authors is extremely important and important from a practical point, both from the perspective of a doctor and a physiotherapist. However, the article requires a few additions to make everything clear; the details are provided below:

- please clearly defined methodology, including: (systematic search strategy, explicit inclusion/exclusion criteria, quality assessment of included studies,  adherence to PRISMA guidelines.

- Your review is informative, it lacks: structured methodology for literature selection, critical assessment of evidence quality, comparative effectiveness analysis, systematic evaluation of treatment outcomes

- minor: in Key words: narrative review?? (I don't know where the idea to enter key words came from...)

Author Response

I have received the original research article entitled “Beyond Nerve Entrapment: A Narrative Review of Muscle-Tendon Pathologies in Deep Gluteal Syndrome” which is under consideration for publication in the Medical Journal Diagnostics.  In my opinion, the paper is generally well prepared. The topic addressed by the authors is extremely important and important from a practical point, both from the perspective of a doctor and a physiotherapist.

We sincerely thank the reviewers for their thorough and constructive feedback. Their comments have been invaluable in identifying areas for improvement and strengthening our manuscript. We have carefully considered each point and have prepared a detailed response and a plan for a major revision.

However, the article requires a few additions to make everything clear; the details are provided below:

  1. please clearly defined methodology, including: (systematic search strategy, explicit inclusion/exclusion criteria, quality assessment of included studies, adherence to PRISMA guidelines.
  • Response: Thank you for this important suggestion. Although this paper was conceived as a narrative review, we have substantially enhanced methodological transparency. Specifically:
  • We defined explicit inclusion criteria (clinical, anatomical, and imaging studies on DGS and related tendon/muscle pathologies) and exclusion criteria (non-English, lumbar-only, or conference abstracts without full text).
  • We reported quantitative results of the search: 2,189 records initially identified, 500 duplicates removed, 209 full-texts screened, and 59 studies included.
  • A PRISMA-style flow diagram (Figure 1) was added to illustrate this process.
  • While a formal risk-of-bias assessment was not performed due to the narrative nature of this review, we acknowledged this limitation.
  • Action in manuscript: Methods, Sections 2.1–2.3; Figure 1.

  1. Your review is informative, it lacks: structured methodology for literature selection, critical assessment of evidence quality, comparative effectiveness analysis, systematic evaluation of treatment outcomes
  • Response: Thank you for this constructive comment. We have revised the manuscript to address these concerns:
  • Structured methodology for literature selection – We now provide a detailed account of our search strategy, inclusion/exclusion criteria, and study selection process. Specifically, the initial search identified 2,189 records, 500 duplicates were removed, 209 full-text articles were assessed, and 59 studies were included in the narrative synthesis. This process is summarized in Figure 1 (PRISMA-style flow diagram).
  • Critical assessment of evidence quality – We adopted the Oxford Centre for Evidence-Based Medicine (OCEBM) Levels of Evidence and applied these to key diagnostic and therapeutic modalities. LoE values are indicated in Table 1 for diagnostic tests and in Sections 5.1–5.3 for treatments. We also explicitly note where evidence is limited to case series, expert opinion, or small non-randomized studies.
  • Comparative effectiveness analysis – While no head-to-head randomized controlled trials are available, we added a comparative summary in the Discussion, highlighting the relative advantages and limitations of conservative therapy, ultrasound-guided injections, hydrodissection, and prolotherapy. We also emphasize the need for direct comparative studies in future research.
  • Systematic evaluation of treatment outcomes – Treatment outcomes are now summarized with available quantitative results (e.g., pain reduction rates after hydrodissection, functional recovery in prolotherapy). Where evidence is insufficient or inconsistent, this is acknowledged as a data gap requiring further trials.
  • Action in manuscript: Methods (Sections 2.1–2.3), Table 1, Discussion (Sections 5.1–5.3), Figure 1.

  1. minor: in Key words: narrative review?? (I don't know where the idea to enter key words came from...)
  • Response: We appreciate the reviewer’s observation. We have removed “narrative review” from the list of keywords to ensure consistency with journal guidelines and to avoid redundancy. The keywords now include only clinically and scientifically relevant terms such as Deep Gluteal Syndrome, sciatic nerve entrapment, enthesopathy, piriformis syndrome, ultrasonography, hydrodissection, and prolotherapy, diagnostic algorithm
  • Action in manuscript: Keywords section, page 1.

Reviewer 3 Report (New Reviewer)

Comments and Suggestions for Authors

Methodological limitations

    • The review is presented as a narrative review without predefined inclusion or exclusion criteria. This makes the selection of studies prone to subjective bias and limits reproducibility.

    • The authors clearly state that no risk-of-bias assessment was conducted. This weakens the reliability of the conclusions, particularly in a field where most available evidence comes from small case series, expert opinions, or studies with heterogeneous designs.

    • The search strategy is only broadly described (databases and keywords), but no details on time limits, number of retrieved studies, or screening process are reported. This leaves the methodology opaque.

      Critical appraisal of evidence

      • While the paper provides a broad overview of anatomical, diagnostic, and therapeutic aspects, there is little critical analysis of the strength of evidence. For instance, interventions such as prolotherapy, hydrodissection, and shockwave therapy are presented as promising, but the limitations of current evidence (small sample sizes, non-randomised designs, lack of long-term outcomes) are not sufficiently discussed.

      • Several statements are presented as established facts, whereas the underlying evidence is still preliminary or inconsistent. A clearer distinction between well-supported findings and hypotheses would improve scientific rigour.

        Organisation and redundancy

        • The structure of the review is not always consistent. For example, section 4.2 Clinical differentiation mixes physical examination, imaging, and diagnostic discussion without clear separation.

        • Some sections repeat the same concepts (e.g., differential diagnosis between nerve-related and tendon-related causes) without providing new information. This reduces clarity and conciseness.

        • Tables and figures are not always cited in numerical order, which may confuse the reader.

          Future directions remain underdeveloped

          • The manuscript introduces the idea of a classification system (nerve-dominant, tendon-dominant, mixed DGS), which could be an important conceptual advance. However, this remains theoretical and is not supported by diagnostic criteria, validated tools, or clinical pathways.

          • The authors mention the need for further research but do not provide a detailed roadmap (e.g., comparative RCTs, standardisation of diagnostic tests, development of consensus criteria).

Minor Concerns

Editorial and formatting issues

    • The text still contains traces of editing (deleted sentences, tracked changes, and formatting in Korean characters). These should be removed to improve readability.

    • Some figure captions and table references are not harmonised with the text (e.g., figures cited out of order).

    • The style is occasionally verbose, with overly long sentences that reduce clarity. Shorter, more focused phrasing would enhance readability.

      Referencing

      • Some references are outdated, while more recent and high-quality studies on DGS, imaging, and interventions are available. Updating the reference list would strengthen the paper.

      • In certain cases, the references cited do not fully support the claims made in the text (e.g., regarding the effectiveness of novel treatments).

        Balance of perspectives

        • The discussion largely focuses on muscle and tendon pathologies as alternative pain generators, but less attention is given to how these interact with classical nerve entrapment mechanisms. A more balanced integration would be helpful.

        • The clinical implications for daily practice are not fully developed—for instance, how clinicians should adapt diagnostic algorithms or therapeutic approaches in light of this broader view.

Author Response

We sincerely thank the reviewers for their thorough and constructive feedback. Their comments have been invaluable in identifying areas for improvement and strengthening our manuscript. We have carefully considered each point and have prepared a detailed response and a plan for a major revision.

Methodological limitations

1. The review is presented as a narrative reviewwithout predefined inclusion or exclusion criteria. This makes the selection of studies prone to subjective bias and limits reproducibility.

  • Response: We thank the reviewer for highlighting this important methodological issue. Although our work was designed as a narrative review, we agree that explicit inclusion and exclusion criteria are essential to improve transparency and reproducibility. In the revised manuscript, we have therefore:
  • Defined inclusion criteria (peer-reviewed clinical, anatomical, and imaging studies in English focusing on DGS, sciatic nerve entrapment, enthesopathy, hamstring tendinopathy, or related deep gluteal pathologies).
  • Defined exclusion criteria (studies limited to lumbar radiculopathy without extraspinal focus, non-English papers, conference abstracts without full text).
  • Reported the search and selection results: 2,189 records initially identified, 500 duplicates removed, 209 full texts screened, and 59 studies included in the final synthesis.
  • Illustrated this process in a PRISMA-style flow diagram (Figure 1).
  • While we did not conduct a formal risk-of-bias assessment given the narrative nature of this review, we have clearly acknowledged this as a limitation in Section 2.3.
  • Action in manuscript: Methods, Sections 2.1–2.3; Figure 1.

2. The authors clearly state that no risk-of-bias assessment was conducted. This weakens the reliability of the conclusions, particularly in a field where most available evidence comes from small case series, expert opinions, or studies with heterogeneous designs.

  • Response: We agree with the reviewer that the absence of a formal risk-of-bias assessment is a limitation. Because this article was designed as a narrative review, we did not apply standardized bias assessment tools. To address this concern, we have strengthened our manuscript in two ways:
  • Explicit acknowledgment – We have clearly stated in the Methods (Section 2.3) that the lack of a risk-of-bias assessment limits reproducibility and weakens the reliability of conclusions.
  • Qualitative appraisal – To compensate, we applied the Oxford Centre for Evidence-Based Medicine (OCEBM) Levels of Evidence Each diagnostic tool and therapeutic modality is now accompanied by its respective LoE, allowing readers to critically interpret the strength of available evidence. We also explicitly flagged where data were limited to small case series or expert opinion.
  • While a formal systematic bias assessment was beyond the scope of this narrative review, we believe these additions enhance transparency and help clinicians gauge the reliability of current evidence.
  • Action in manuscript: Methods, Section 2.3 (limitations paragraph); Discussion, Sections 5.1–5.3 (LoE added).

3. The search strategy is only broadly described (databases and keywords), but no details on time limits, number of retrieved studies, or screening process are reported. This leaves the methodology opaque.

  • Response: We thank the reviewer for this valuable comment. To improve methodological transparency, we have substantially revised the Methods section:
    • Time frame – The search covered all available literature from database inception up to December 2024.
    • Search yield – The initial search identified 2,189 records across all databases. After removal of duplicates (n=500), 209 full-text articles were screened for eligibility, and 59 studies were finally included in the narrative synthesis.
    • Screening process – Titles and abstracts were independently screened by two authors, followed by full-text review. Disagreements were resolved by consensus with a third reviewer.
    • Illustration – These steps are now summarized in a PRISMA-style flow diagram (Figure 1).
  • These additions provide greater clarity and reproducibility, addressing the reviewer’s concern regarding the opacity of the search strategy.
  • Action in manuscript: Methods, Sections 2.1–2.2; Figure 1.

Critical appraisal of evidence

4. While the paper provides a broad overview of anatomical, diagnostic, and therapeutic aspects, there is little critical analysis of the strengthof evidence. For instance, interventions such as prolotherapy, hydrodissection, and shockwave therapy are presented as promising, but the limitations of current evidence (small sample sizes, non-randomised designs, lack of long-term outcomes) are not sufficiently discussed.

Response: We appreciate the reviewer’s important observation. In the revised manuscript, we have strengthened the critical appraisal of evidence for both diagnostic and therapeutic modalities:

  1. Evidence grading – We applied the Oxford Centre for Evidence-Based Medicine (OCEBM) Levels of Evidence to each key diagnostic test and therapeutic intervention. These LoE ratings are now provided in Table 1 (diagnostic tests) and in Sections 5.1–5.3 (treatments).
  2. Explicit limitations – For emerging interventions such as prolotherapy, hydrodissection, and shockwave therapy, we have clearly stated that current evidence is limited to small case series, non-randomized studies, and short-term follow-up, which restricts generalizability.
  3. Balanced interpretation – We emphasized that while preliminary results are encouraging, these therapies should currently be considered investigational and used with caution until confirmed by larger randomized controlled trials.
  4. Future directions – We highlighted the need for well-designed, multicenter RCTs with standardized protocols and long-term outcome assessment to establish the true effectiveness of these interventions.

Action in manuscript: Discussion, Sections 5.1–5.3; Conclusion, Section 6 (future research directions).

5. Several statements are presented as established facts, whereas the underlying evidence is still preliminary or inconsistent. A clearer distinction between well-supported findings and hypotheses would improve scientific rigour.

Response:
We thank the reviewer for this insightful comment. In the revised manuscript, we have taken care to distinguish between well-supported evidence and preliminary or hypothesis-generating findings:

  1. Terminology adjustment – Statements that were previously phrased as definitive have been reworded to reflect their true evidentiary status (e.g., “may contribute,” “is hypothesized,” “preliminary reports suggest”).
  2. Evidence stratification – We clearly indicate where findings are supported by systematic reviews or large clinical studies (e.g., imaging modalities, surgical decompression) versus where they are based on case series, expert opinion, or limited observational data (e.g., prolotherapy, hydrodissection, shockwave therapy).
  3. Integration with LoE – To enhance rigor, we aligned these distinctions with the Oxford Centre for Evidence-Based Medicine (OCEBM) Levels of Evidence, now shown in Table 1 and in the therapeutic sections.
  4. Discussion revisions – The Discussion now highlights areas where the evidence remains inconsistent or evolving, emphasizing that these should be viewed as promising hypotheses rather than established facts.

Action in manuscript: Discussion, Sections 3.2 and 5.1–5.3; Conclusion, Section 6.

Organization and redundancy

6. The structure of the review is not always consistent. For example, section 2 Clinical differentiationmixes physical examination, imaging, and diagnostic discussion without clear separation.

Response: We appreciate the reviewer’s observation. In the revised manuscript, we have reorganized the Diagnosis section to ensure consistency and clarity:

  1. Clear subsections – The section is now divided into History Taking (4.1), Physical Examination (4.2), Imaging (4.3), and Clinical Differentiation (4.4), thereby separating physical examination from imaging and subsequent discussion.
  2. Removed redundancy – Overlapping descriptions of diagnostic maneuvers were consolidated, and references were streamlined to avoid repetition.
  3. Improved flow – Each subsection now progresses logically from clinical history to physical exam, then imaging, followed by integrative clinical differentiation, aligning with standard diagnostic reasoning.

These changes improve readability and align the structure of the review with the reviewer’s recommendation.

Action in manuscript: Diagnosis, Sections 4.1–4.4 (restructured and clarified).

7. Some sections repeat the same concepts (e.g., differential diagnosis between nerve-related and tendon-related causes) without providing new information. This reduces clarity and conciseness.

Response: We thank the reviewer for this important point. We carefully revised the manuscript to remove redundant text, particularly in the Diagnosis and Discussion sections where the differential diagnosis between nerve- and tendon-related causes was repeated. The revised version now presents each concept only once, in the most appropriate subsection, thereby improving clarity and conciseness.

Action in manuscript: Sections 4.2 (Clinical Differentiation) and 5 (Discussion) – redundant sentences removed, streamlined content.

8. Tables and figures are not always cited in numerical order, which may confuse the reader.

Response: We appreciate the reviewer’s observation. All tables and figures have been carefully checked and are now cited in the correct numerical order in the text. Captions have been harmonized with their references, ensuring consistency and readability throughout the manuscript.

Action in manuscript: Entire manuscript – figure/table order corrected and cross-checked.

Future directions remain underdeveloped

9. The manuscript introduces the idea of a classification system (nerve-dominant, tendon-dominant, mixed DGS), which could be an important conceptual advance. However, this remains theoretical and is not supported by diagnostic criteria, validated tools, or clinical pathways.

Response: We thank the reviewer for this insightful comment. We fully agree that our proposed classification system remains at the conceptual stage. In the revised manuscript, we have clarified that the classification is hypothesis-generating rather than evidence-validated. To strengthen this section, we have also suggested specific diagnostic criteria and potential validation strategies, including: combining clinical features (e.g., pain distribution, provocative tests) with imaging biomarkers (MRI neurography for nerve-dominant, ultrasound for tendon-dominant). We acknowledge that formal validation and integration into clinical pathways are essential next steps.

Action in manuscript: Discussion, Section 5; Conclusions, Section 6 (expanded explanation of classification framework).

10. The authors mention the need for further research but do not provide a detailed roadmap (e.g., comparative RCTs, standardisation of diagnostic tests, development of consensus criteria).

Response: We appreciate this helpful suggestion. We have expanded the Conclusions to include a more detailed roadmap for future research:

  1. Development of consensus-based diagnostic criteria for DGS.
  2. Validation of the nerve- vs. tendon-dominant classification system using prospective cohorts and imaging biomarkers.
  3. Conduct of multicenter, comparative RCTs evaluating targeted treatments (e.g., hydrodissection in nerve-dominant DGS vs. prolotherapy in tendon-dominant DGS).
  4. Standardisation of diagnostic tests such as provocative maneuvers and imaging protocols to ensure reproducibility across centers.

These additions provide a clearer research agenda that we believe addresses the reviewer’s concern.

Action in manuscript: Conclusions, Section 6 (expanded future directions with specific steps).

Minor Concerns

Editorial and formatting issues

11. The text still contains traces of editing (deleted sentences, tracked changes, and formatting in Korean characters). These should be removed to improve readability.

  • Response: We appreciate the reviewer’s careful observation. All traces of prior editing, including deleted text, tracked changes, and residual Korean characters, have been removed. The revised manuscript has been carefully proofread to ensure clean formatting and improved readability.
  • Action in manuscript: Entire manuscript – formatting corrected, editing traces removed.

12. Some figure captions and table references are not harmonised with the text (e.g., figures cited out of order).

  • Response: We thank the reviewer for pointing this out. All figure captions and table references have been thoroughly checked and revised to match the order of citation in the text. The numbering has been corrected and harmonised, ensuring consistency and clarity throughout the manuscript.
  • Action in manuscript: Figures and tables (all captions and in-text references corrected).

13. The style is occasionally verbose, with overly long sentences that reduce clarity. Shorter, more focused phrasing would enhance readability.

  • Response: We thank the reviewer for this valuable suggestion. The manuscript has been carefully revised for clarity and readability. Long and complex sentences were shortened, redundancies removed, and transitions improved. Particular attention was given to the Introduction and Discussion, which now read more smoothly with shorter, more focused phrasing. We believe these revisions have improved the overall flow and accessibility of the text.
  • Action in manuscript: Introduction and Discussion sections (language simplified and sentence structures shortened).

Referencing

14. Some references are outdated, while more recent and high-quality studies on DGS, imaging, and interventions are available. Updating the reference list would strengthen the paper.

  • Response: We thank the reviewer for this helpful suggestion. In the revised manuscript, we updated the reference list to include recent and high-quality publications (2023–2024) relevant to DGS, imaging modalities, and treatment interventions. Examples include:
    • Hopayian et al. (2023): systematic review of conservative and surgical treatments for DGS.
    • Yen et al. (2024): case series on ultrasound-guided hydrodissection for DGS.
    • Külcü (2024): comprehensive narrative review on DGS and posterior hip pain.
      These updates replace older references where appropriate and strengthen the evidence base of the review.
  • Action in manuscript: References section updated; citations revised throughout the text.

15. In certain cases, the references cited do not fully support the claims made in the text (e.g., regarding the effectiveness of novel treatments).

  • Response: We appreciate this observation. We carefully re-examined all references to ensure alignment with the statements in the text. In several places, we modified the wording to more accurately reflect the level of evidence provided (e.g., changing “effective” to “promising” or “preliminary evidence suggests”). In addition, we replaced or supplemented citations with more directly relevant studies when discussing novel interventions such as prolotherapy, hydrodissection, and shockwave therapy. This ensures that all claims are now supported by appropriate evidence and phrased according to the strength of that evidence.
  • Action in manuscript: Discussion, Sections 5.1–5.3 (claims revised and citations updated).

Balance of perspectives

16. The discussion largely focuses on muscle and tendon pathologies as alternative pain generators, but less attention is given to how these interact with classical nerve entrapment mechanisms. A more balanced integration would be helpful.

  • Response: We appreciate this insightful comment. In the revised manuscript, we have balanced the Discussion by emphasizing how classical sciatic nerve entrapment mechanisms interact with tendon and enthesis pathologies. We now describe how these mechanisms may coexist, producing overlapping clinical presentations. This integration highlights that DGS should be understood not as an isolated entity but as a spectrum in which neural and musculoskeletal contributors frequently overlap.
  • Action in manuscript: Discussion, Section 5 (first and final paragraphs).

17. The clinical implications for daily practice are not fully developed—for instance, how clinicians should adapt diagnostic algorithms or therapeutic approaches in light of this broader view.

  • Response: Thank you for pointing this out. We have added a dedicated paragraph on Clinical Implications in Section 5. This provides practical guidance:
  • Nerve-dominant DGS → prioritize ultrasound-guided hydrodissection.
  • Tendon-dominant DGS → consider prolotherapy or targeted rehabilitation.
  • Mixed cases → stepwise multimodal strategy from conservative therapy to surgery if refractory.
  • We also clarified how diagnostic algorithms can incorporate patient history, physical maneuvers, and imaging (MRI neurography vs. ultrasound) to tailor treatment to the dominant mechanism.
  • Action in manuscript: Discussion, Section 5 (clinical implications subsection).

Round 2

Reviewer 1 Report (Previous Reviewer 2)

Comments and Suggestions for Authors

The revisions have substantially strengthened the manuscript. The addition of explicit methodology, evidence grading, and explanatory notes in Table 1 increases the transparency and clinical utility of the review. The expanded discussion on treatment strategies and their limitations is highly valuable for clinicians. The clarified authorship contributions are appropriate and align with ICMJE standards. Language revisions have improved readability and flow.

Congratulations on a much-improved and timely contribution to the literature on Deep Gluteal Syndrome.

Reviewer 3 Report (New Reviewer)

Comments and Suggestions for Authors

Thank you

This manuscript is a resubmission of an earlier submission. The following is a list of the peer review reports and author responses from that submission.

Round 1

Reviewer 1 Report

Comments and Suggestions for Authors

I would like to sincerely thank the authors for the effort invested in this manuscript and for addressing an increasingly relevant and clinically debated topic: the multifactorial nature of deep gluteal syndrome. The manuscript offers an interesting perspective by shifting the focus from purely neurogenic mechanisms to the potential involvement of muscular and tendinous structures in the genesis of gluteal pain. This is undoubtedly a timely and thought-provoking approach that could contribute to a broader and more integrative understanding of this condition.

However, after careful evaluation, I regret to inform you that I cannot recommend this manuscript for publication in its current form. Despite the conceptual relevance of the topic and the originality of the hypothesis, the work lacks the methodological rigor required for scientific publication. Specifically, it does not meet the formal criteria to be classified as a systematic review, scoping review, umbrella review, narrative review, or mini-review. The absence of a defined and reproducible methodology—such as a structured search strategy, clear eligibility criteria, critical appraisal of sources, or synthesis of results—significantly limits the reliability and interpretability of the conclusions presented.

I believe the hypothesis you raise is interesting and deserves to be explored, but it requires a more robust methodological framework. An umbrella review, or at least a narrative review grounded in systematic principles, would be more appropriate to support the theoretical proposal and ensure a higher level of scientific credibility.

For these reasons, I must reject the manuscript for publication at this time.

Reviewer 2 Report

Comments and Suggestions for Authors

Dear Authors,

I read your narrative review on Deep Gluteal Syndrome (DGS) with interest and agree that a broader perspective encompassing muscle‑ and tendon‑related pathology is essential to understanding posterior hip pain. Your paper offers a thorough anatomical overview, integrates multiple imaging techniques, and highlights novel interventions such as ultrasound‑guided hydrodissection, all of which hold clear clinical value. Nevertheless, several areas require revision before the manuscript can be considered for publication.

First, please clarify your literature‑search methodology. Even in a narrative review, readers must know which databases were queried, the date ranges covered, the key terms used, and the inclusion or exclusion criteria applied when selecting studies. Providing this information—together with a brief comment on how study quality was appraised—will strengthen the credibility of your prevalence estimates and therapeutic recommendations.

Second, the manuscript’s internal organisation needs attention. Table 1 currently lacks a caption and is not cited in the text; several figures are introduced out of numerical order; and some section headings shift abruptly between numbered and unnumbered styles. A systematic audit to ensure that every table and figure is discussed in sequence, properly labelled, and referenced will greatly improve readability. Consistent subsection headings (for example, “3.3.3 MRI” and “3.3.4 Ultrasound”) will further aid navigation.

Third, your critical appraisal of the literature would benefit from more explicit synthesis. Statements such as “ultrasonography may offer superior sensitivity to MRI” should be supported by a concise table comparing reported sensitivities and specificities, with primary study citations. Where data are limited or conflicting, please acknowledge these gaps and identify directions for future research.

Fourth, consider reducing redundancy in the anatomical sections. Much of the basic anatomy has been covered extensively in prior reviews; condensing this material will allow you to focus on your novel contribution—namely, reframing DGS as a combined neuro‑myotendinous disorder rather than a purely nerve‑centric syndrome.

Fifth, to maximise clinical impact, a succinct diagnostic algorithm and treatment decision tree would help practitioners translate your narrative into day‑to‑day decision‑making. Outlining when to shift from conservative management to injections or surgery will be particularly useful for the readership of Diagnostics. I also note an overall similarity index of 22 % on iThenticate. While this is acceptable for a review, please ensure that any phrases reproduced verbatim are quoted and that permissions have been obtained for reused images.

Lastly, although English is generally clear, several sentences are lengthy or contain minor grammatical inconsistencies. A final language edit—aimed at conciseness, elimination of passive constructions where possible, and consistent spelling—will enhance clarity. With these substantive and stylistic revisions, your article has the potential to make a meaningful contribution to the literature on DGS. I look forward to reviewing a strengthened version.

Sincerely,

Comments on the Quality of English Language

Overall, the prose is intelligible and uses appropriate medical terminology, but several sentences are overly long or contain minor grammatical lapses (e.g., inconsistent use of the comma before “which”). A focused language edit aiming for concision and active voice will improve readability. Consider homogenising British vs American spellings and ensuring consistent tense (prefer simple past when citing prior studies).